# Can 'On-Farm' Seed Priming and Chitosan Seed Treatments Induce Host Defences in Winter Barley (*Hordeum vulgare* L.) under Field Conditions?

Javier Carrillo-Reche [1],*, Adrian C. Newton [2], Francesc Ferrando-Molina [1,2] and Richard S. Quilliam [1]

1   Biological and Environmental Sciences, Faculty of Natural Sciences, University of Stirling, Stirling FK9 4LA, UK; francesc.ferrandomolina1@stir.ac.uk (F.F.-M.); richard.quilliam@stir.ac.uk (R.S.Q.)
2   Ecological Sciences, The James Hutton Institute, Dundee DD2 5DA, UK; adrian.newton@hutton.ac.uk
*   Correspondence: javier.carrilloreche1@stir.ac.uk

**Abstract:** Enhancing host defences through induced resistance, disease tolerance, and/or escape, in combination with current disease management regimes may be a valuable strategy to reduce pesticide use. Since both 'on-farm' seed priming (OSP) and chitosan priming (CHP) have been reported to confer varying levels of host defence, this study sought to investigate their potential to deliver disease control as a strategy for sustainable management of foliar pathogens in winter barley. Field experiments were conducted to determine the effects of OSP and CHP at two different field sites using three different cultivars under fungicide/non-fungicide regimes. Overall, no evidence was found to suggest that CHP or OSP can induce effective resistance in temperate field conditions. However, these field trials enabled the identification of candidate traits to deliver disease tolerance (and escape) for the primary and secondary spread of powdery mildew, i.e., large canopies and rapid stem elongation respectively. Thus, these seed treatments may deliver disease tolerance and escape traits, but these benefits are dependent upon successful establishment and vigour first. The integration of seed treatments into sustainable crop protection may be better undertaken with spring crops or in semi-arid agriculture where the added vigour at emergence can help compensate for negative environmental interactions.

**Keywords:** canopy structure; elicitor; foliar disease; induced resistance; tolerance trait; grain yield



## 1. Introduction

Plant host defence against pathogens and parasites involves three elements: (1) 'resistance', which is the capacity of a crop to eliminate or limit pests and pathogens by genetic and molecular mechanisms; (2) 'tolerance', which is the capacity of the plant to maintain performance in the presence of disease symptoms; and (3) 'escape', which is the ability to restrict the dispersal of inocula within the canopy and hence the spread of the disease [1,2].

A number of natural and synthetic substances have the potential to induce host resistance; these so-called plant defence elicitors include chitosan, which acts as a priming stimulus for systemic resistance by mimicking pathogen-associated molecular pattern molecules (PAMPs) [3,4]. Chitosan can induce resistance in crop plants against a wide range of pathogens including via its direct application to seeds. For example, seeds from tomato, pearl millet, and wheat immersed in a chitosan solution gained subsequent protection against *Fusarium oxysporum*, *Sclerospora graminicola*, and *F. graminearum* respectively, through the accumulation of defence-related secondary metabolites, e.g., beta-1,3 glucanase and ferulic acid [5–7]. Chitosan can also confer other physiological effects that may result in varying degrees of disease tolerance. For example, chitosan application can enhance seedling vigour, resulting in increased net photosynthetic rate and a larger canopy [8,9], both of which are traits that can lead to the tolerance of foliar diseases in cereal crops [10]. However, field-scale data quantifying the effects of chitosan seed treatments on defence responses and their potential implications in yield are lacking [11].

A low-cost seed treatment technique called 'on-farm' seed priming has been recognised as an effective approach to alleviate adverse seedbed conditions, such as soil crusting and limited soil moisture. These often limit yield potential in semi-arid areas of developing countries [12]. Yield improvements delivered by 'on-farm' seed priming often exceed the expected gain due to better establishment; i.e., there is an added agronomic advantage [12]. Varying degrees of increased host defence expression have been proposed as supplementary mechanisms [13]. 'On-farm' seed priming consists of anaerobically soaking seeds in water for a predetermined duration before sowing [14]. The hypoxic conditions, together with membrane damage caused by rapid uncontrolled imbibition, can trigger an accumulation of phytohormones associated with induced resistance. Upon pathogen attack, these could accelerate and strengthen defence responses [13]. For example, this was the case for downy mildew (causal agent *Sclerospora graminicola*) in pearl millet, where a 20% decrease in infection was associated with induced resistance responses following 'on-farm' priming of seeds [13]. Physiological and phenological effects derived from enhanced vigour and hastened metabolism following 'on-farm' seed priming are also involved with different forms of disease tolerance and/or escape for a number of tropical crops. A considerably decreased severity of the symptoms caused by mungbean yellow mosaic virus (MYMV) was attributed to the improved vigour and state of readiness of the plant to defend itself (i.e., plant 'tolerance') [15]. A rapid emergence of crops following 'on-farm' seed priming reduced the size of the 'infection window' available to soil-borne diseases such as collar rot (*Sclerotium rolfsii*) and *Fusarium* wilt in chickpea [16]. In addition, the decreased time to maturity reduced the exposure to late-season pests in maize (i.e., 'escape') [17]. However, so far, 'on-farm' seed priming has not been investigated for arable crops in temperate agricultural systems, and with the increasing pressure to reduce chemical use (including those used in fungicide seed treatments), non-chemical treatments are set to increasingly gain importance in more agroecological cropping schemes [18].

Effective control of diseases solely through induced resistance, tolerance, and/or escape mechanisms is unlikely. However, unlike fungicides or genetically mediated resistance, these strategies are broad-spectrum and so do not generate strong pathogen-specific selection pressure. Enhancing host defences, in combination with current disease management regimes, may be a valuable strategy to reduce pesticide use and provide durable disease control in integrated pest management (IPM) programmes of cereal grains [1]. In barley, it is especially important to protect crops from early epidemics during the vegetative growth, as yield largely relies on maximised tiller production and survival (sink limited) [1,10,19]. Thus, forming a well-sized canopy early in the season can be a candidate trait for tolerating foliar diseases, as it would minimise the effects of disease on growth [10,20]. Traits such as increased height and rapid stem elongation are known to be useful traits for reducing the spread of disease to upper leaves [1,21]. Such an 'escape' mechanism may be effective against temperate crop diseases, e.g., the splash-dispersed rhynchosporium (*Rhynchosporium commune*), by reducing early infection and subsequent epidemics. All things considered, the enhanced vigour commonly conferred by seed treatments may be particularly valuable in winter barley (more routinely exposed to overwintering pathogens than spring barley) to retain tillers that might otherwise be lost to disease [22]. However, evaluation of individual tolerance traits alone may not bring a complete insight, as tolerance is the result of multiple traits operating at organ, plant, and crop level [2]. Quantification of tolerance as the slope of a relationship of yield on healthy area duration (HAD, the area under the green leaf lamina area progress curve) has been used to more holistically assess tolerance in wheat, which may also be applicable to barley [2,23,24].

Therefore, the overall aim of this study was to investigate the potential of 'on-farm' seed priming and chitosan-based seed treatments to deliver disease control as a strategy for the sustainable management of winter barley pathogens. Specifically, our objectives were to test the hypotheses that 'on-farm' seed priming and chitosan seed treatment can achieve the following: (a) induce disease resistance; (b) confer disease tolerance and/or an

escape response; and (c) increase crop yields in a temperate field-scale agricultural context. Additionally, we examined canopy size before stem elongation and stem elongation rate as candidate traits to deliver tolerance and escape against foliar diseases, and we implement yield-HAD slopes to estimate the overall tolerance in barley.

## 2. Materials and Methods

### 2.1. Plant Material and Preparation of Seed Treatments

Three winter barley genotypes with differential responses to common foliar diseases—according to the Agriculture and Horticulture Development Board (AHDB) Recommended Lists for cereals and oilseeds [25]—were selected (Table 1). Seed treatments consisted of an 'on-farm' seed priming treatment (OSP), chitosan (CHP) applied as ChitoPlant® (ChiPro GmbH, Bremen, Germany) at a concentration of 0.5 g L$^{-1}$ based on previous findings [8], and a non-primed control (NP), which consisted of dry seeds. Preliminary tests were carried out to determine the optimal 'on-farm' seed priming duration for each cultivar as described in Carrillo-Reche et al. [26]. The optimal priming durations were 20, 24, and 28 h for SY Venture, KWS Tower, and KWS Cassia, respectively (see Figure S1).

**Table 1.** Details of cultivars used in both growth trials.

| Cultivar | Date Listed | Type | Resistance Mildew [1] | Resistance Rhynchosporium [1] |
|---|---|---|---|---|
| SY Venture | 2012 | Two-row malting | 6 | 4 |
| KWS Cassia | 2010 | Two-row feed | 4 | 4 |
| KWS Tower | 2014 | Two-row feed | 5 | 5 |

[1] Resistance ratings according to the 'AHDB Recommended Lists for cereals and oilseeds 2018/19' [25] on a scale of 1–9, with a higher value indicating a higher resistance.

Approximately 13,400 seeds of each cultivar (calculated by weight from the thousand grain weight) were added to 5 L plastic buckets containing either distilled water or 0.5 g L$^{-1}$ chitosan solution (1:5 (*w/v*) ratio). All buckets were incubated at 20 °C for the corresponding optimal priming durations for each cultivar, or 15 min for CHP treatments. After soaking, OSP seeds were oven-dried at 50 °C until moisture content was reduced to 27–31% (sufficiently dry to avoid clumping within the seed drill pipes). The moisture content of the NP and CHP treatments ranged from 12 to 16%. Subsequently, seed were re-weighed and split into twelve equal weight portions (which provided the twelve replicates for each cultivar × seed treatment combination) and packed in paper envelopes prior to sowing.

### 2.2. Field Sites, Experimental Design, and Crop Husbandry

Winter barley trials were conducted at two sites near Dundee, UK (Table 2). The first site, Hutchens at Balruddery, was selected as a representative site for growing barley within a rotation. The second site, East Loan at Mylnefield, has had barley repeatedly cultivated as a monoculture and has been used as a disease nursery for cultivar testing for over 30 years.

**Table 2.** Conditions of both growth trials during the 2019–2020 season.

| Site | Sowing Date | Latitude, Longitude | Elevation (m) | Soil Texture | Previous Crops | Harvest Date |
|---|---|---|---|---|---|---|
| Balruddery-Hutchens | 17 October | 56°29′03.5″ N 3°06′34.4″ W | 118 | Sandy loam | Barley (2017), Peas (2018) | 31 July |
| Mylnefield-East Loan | 29 October | 56°27′21.4″ N 3°04′25.2″ W | 13 | Sandy loam | Barley since 1986 | 2 August |

At both sites, the experimental design consisted of two crop protection treatments, either no fungicide (F0) or fungicide (F1) applied alternately per column; and three replicates (Figure 1). Fungicides (Table 3) were applied according to standard pesticide protocols with a hand-pump rucksack. Weeds were controlled with pre-emergence herbicides Pincer®

(Agform, Wickham, UK) and PicoMax® (BASF, Cheadle, UK) at 0.6 and 3.0 L ha$^{-1}$ respectively. Adjoining guards of barley surrounding each column were sown to act as a buffer for the fungicide applications and to reduce potential edge effects. Each column contained 18 plots and was split into two sub-reps with the nine cultivar × seed treatment combinations randomised within each sub-rep. Thus, each fungicide × cultivar × seed treatment combination comprised six replicates.

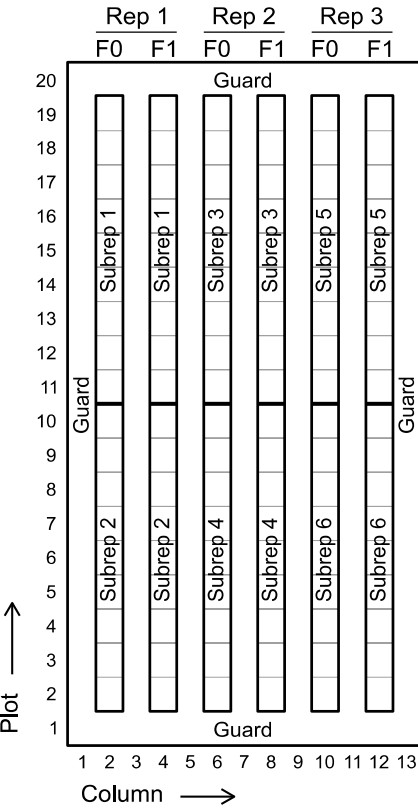

**Figure 1.** Experimental field setup at both sites. Whole plots were arranged along columns and sub-plots by rows, with guards in the middle of the whole plots and sub-plots. Fungicide was applied alternately per column (either none, F0; or full treatment, F1) and sub-replicated in the same column. Each sub-replicate contained nine plots where cultivar × seed treatment combinations were randomised.

**Table 3.** Fungicide programme and active substances.

| Treatment | Commercial Product | Active Ingredient | Rate (L ha$^{-1}$) | GS Applied [1] |
|-----------|--------------------|-------------------|--------------------|----------------|
| T0 | Proline | Prothioconazole | 0.5 | GS 30 |
|    | Corbel | Fenpropimorph | 0.5 | |
| T1 | Siltra Xpro | Bixafen and prothioconazole | 0.6 | GS 31–32 |
|    | Rover 500 | Chlorothalonil | 1 | |
|    | Vegas | Cyflufenamid | 0.3 | |
| T2 | Tucana | Pyraclostrobin | 1 | GS 49 |
|    | Imprex | Fluxapyroxad | 2 | |
|    | Joules | Chlorothalonil | 1 | |
|    | Proline | Prothioconazole | 0.3 | |

[1] There were 19 days between T0 and T1 application and 29 days between T1 and T2 application at both sites. Specific timing of applications can be found in Figure 2.

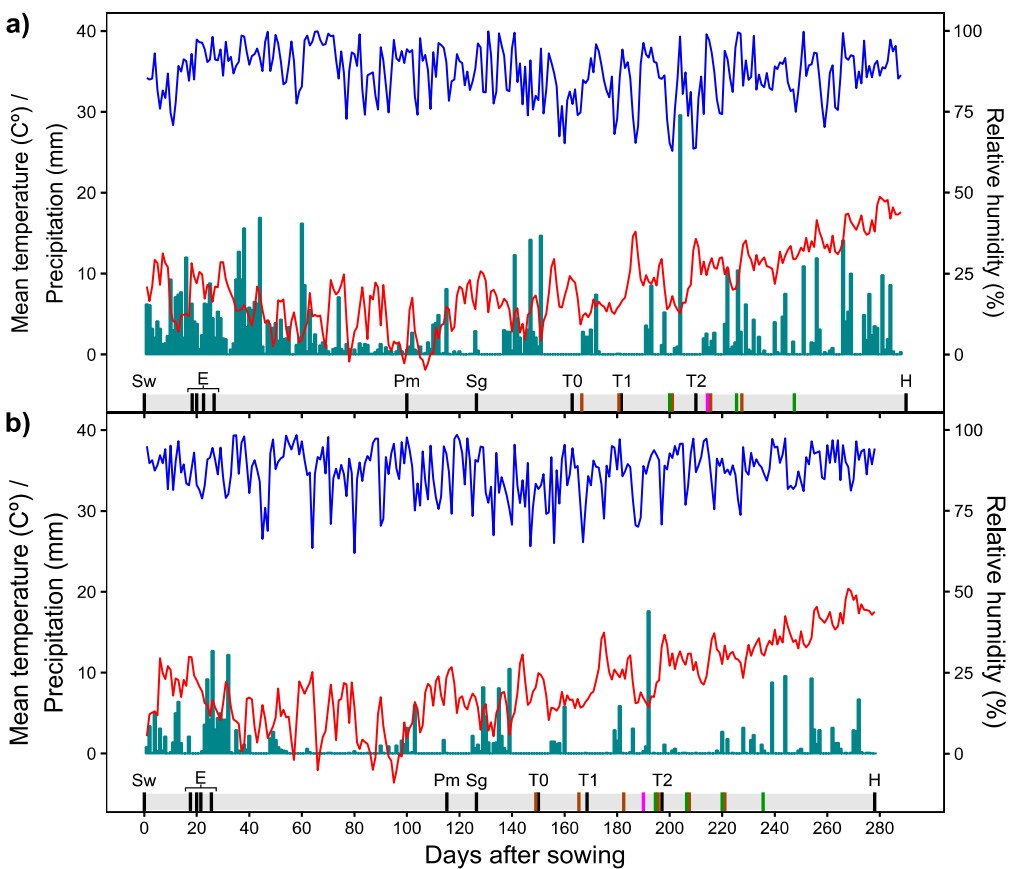

**Figure 2.** Climatic conditions and timing of key activities during the growing season at (**a**) Balruddery and (**b**) Mylnefield. Daily mean temperature (red lines), daily precipitation (green bars), and daily mean relative humidity (blue lines). Black ticks across the shaded strip within the plots represent events of sowing (Sw), emergence counts (E), first powdery mildew pustules appearance (Pm), segmentation images acquisition (Sg), Tx applications (T0–2), and harvest (H). The green ticks represent an image acquisition event for canopy green cover, the brown ticks represent a disease score event, and the pink tick represents when 50% of the stems showed visible awns (GS49).

Plots were sown with an eight-row Hege plot drill (1.55 × 2.00 m) at 360 seed m$^{-2}$ together with a seedbed application of 350 kg ha$^{-1}$ of 0:20:30 nitrogen/phosphorus/potassium (N:P:K). Approximately, a total of 340 kg ha$^{-1}$ of 29:0:0 (7 sulphate [SO$_4$]) was applied at each site. At Balruddery, a half dose was applied in March and the other half was applied in April, whereas a full dose was applied in March at Mylnefield. This was done because an irrigation system was installed in Mylnefield at the beginning of April, which restricted fertiliser application operations. The purpose of the irrigation system was to promote *Rhynchosporium commune* spore dispersal by simulating rain splash via the overhead sprinklers (Rightrain, Ringwood, UK), which were distributed across the experimental field. Irrigation was provided from developmental stage GS31 to 71 and consisted of applications of approximately 15 mm of water three times a week.

### 2.3. In-Field Imaging

#### 2.3.1. Image Collection

Zenithal images of each plot were collected from the stage of emergence of the first seedlings to approximately stage GS71–75 (specific timings of image acquisition are shown in Figure 2). Images were taken 80 cm above the canopy with a Canon EOS 1200D digital camera (Canon, Tokyo, Japan). Where possible, images were taken in a period spanning solar noon (10:00–14:00), particularly on overcast days for consistent light quality. The camera was held parallel to the ground with a monopod and focused near the central area of the plot. The camera was set at 18 mm focal length, automatic aperture with no

flash, and 1/250 shutter speed. The images were stored as JPEG with native resolution of 3456 × 5184 pixels. Prior to the first images being collected, a 1 m section, parallel to the row orientation, was delimited by placing two sticks on the soil between the central rows of each plot. This allowed posterior conversion of pixels to $m^2$ as the long side of the picture (5184 pixels) captured the two sticks at the extremes of the picture (approximately equivalent to 1 m).

### 2.3.2. Image Processing for Emergence Counts

An image capturing the delimited area per plot was used for seedling counts (Figure S2a for illustration), and emergence counts in the same section of the plot in each visit. Seedlings at both side rows of the marked section were counted with a cell counter plugin and zoomed 50× in FIJI software (version 2.0.0-rc-49/1.52s) [27] (Figure S2b for illustration). Images for emergence counts were taken every 2–3 days from the appearance of the first emerged seedlings until it was considered that emergence had reached its end, i.e., when count numbers from the last visit coincided with the counts from the penultimate visit.

### 2.3.3. Image Processing for Leaf Area Index and Percentage of Senescent Tissue Estimation at Advanced Tillering

A single image per plot capturing the delimited area was taken to evaluate early vigour and the severity of an early powdery mildew epidemic at the end of advanced tillering. Leaf area index (LAI) equates to ground cover as plants have not yet gone through stem extension [28]. The timing of image acquisition was at 23 and 13 days after the first observation of disease symptoms at Balruddery and Mylnefield, respectively, and 35 and 23 days before the T0 fungicide application, respectively. To facilitate image segmentation, image acquisition was carried out on a cloudy day to avoid overly bright leaves and several hours after a rain event whilst the soil was still moist, which improved the colour contrast between the green shoot and the soil. Segmentation of soil, green plant tissue, and senescent tissue was performed using FIJI software (Figure S3 for illustration). In brief, pixels within each picture were automatically classified into two clusters depending on their distance to a cluster centroid generated by the k-means++ algorithm using the k-means Clustering plugin (https://github.com/ij-plugins/ijp-toolkit/wiki/k%E2%80%90means-Clustering, accessed on 15 March 2020) in FIJI. This roughly classifies pictures into two layers containing dark/brown (attributable to soil), green, and yellow/light brown pixels (attributable to plant tissue). The layer corresponding to plant tissue was retained, and most of the stones and small particles within the area were eliminated, setting a threshold for particles with high circularity. Subsequently, the resultant RGB image was converted to CIELab colour space (Commission Internationale de l'Eclairage, L* lightness, a* green–red component, b* blue–yellow component) to more finely classify pixels by colour thresholding. Pixels from 0 to 255, 0 to 105, and 120 to 255 degrees for the channels L*, a*, and b* were considered greenish and from 0 to 255, 106 to 135, and 120 to 255 degrees for the channels L*, a*, and b* were considered yellow. At least ten randomly selected images per site were visually inspected to verify the quality of the segmentation before bulk processing. LAI cover was calculated as the sum of green pixels and yellow pixels and converted to $m^2$ being expressed as $m^2$ of LAI $m^{-2}$ of soil. Percentage of senescent tissue (PST) was calculated from the proportion of yellow pixels in the leaf area index.

### 2.3.4. Image Processing for Canopy Green Cover

Two images of each plot were taken that targeted the central rows of the plot, but not necessarily from the delimited area, from plants at stages GS41 to GS71–75 approximately every two weeks. Canopy green area was calculated using CerealScanner plugin ([29]; https://integrativecropecophysiology.com/software-development/cerealscanner/, accessed on 28 May 2020), in FIJI, which is a specialist plugin for the characterisation of canopy growth in cereals [30].

## 2.4. In-Field Measurements

### 2.4.1. Disease Severity

The disease severity of powdery mildew and rhynchosporium was scored on a continuous scale (0–100%) at plot level following the 'AHDB Cereal trials protocol' [31] from GS30 (approximately when T0 was applied) until the distinction between chlorotic and senescent tissue was no longer possible (approximately after GS69). Disease scores were carried out approximately every two weeks.

### 2.4.2. Height and Maturity

Crop height was measured after visually determining the most representative part of the average plot height at stages GS31, GS33, GS49, and GS71. At Balruddery, only the GS71 measurement (final height) was taken. Height was measured from the ground level to the base of the highest fully expanded leaf ligule or, after ear emergence, to the base of the highest ear. The number of days from sowing to GS49 (when approximately 50% of the stems showed visible awns) was recorded for each plot as an estimate of time to crop maturity.

## 2.5. Yield and Grain Quality

Plots were harvested at maturity with a Wintersteiger Plot Combine and dried to a constant moisture. Grain was weighed after being passed through a 2.5 mm sieve, for elimination of remaining awns and small or broken grain. A subsample of cleaned grain was used to determine grain nitrogen concentration (GN), and moisture content determined by using a calibrated near-infrared grain analyser (Infratec 1241, FOSS, Sweden). Thousand grain weight (TGW) was calculated using a MARVIN Seed Analyser (GTA Sensorik, Neubrandenburg, Germany). Then, the grain weight of each plot was adjusted to 85% dry matter to obtain grain yield (GY) and grain number (G no.) calculated from the GY and TGW.

## 2.6. Meteorological Conditions

Mean temperature, accumulated precipitation, and relative humidity data were collected by an automated meteorological station situated at a maximum distance of 300 m from the experimental area (Figure 2). Balruddery weather data were supplied by the Natural Environment Research Council through the COSMOS-UK project (https://cosmos.ceh.ac.uk/, accessed on 21 November 2019) and Mylnefield weather data were supplied by the James Hutton Institute.

## 2.7. Data Analysis

Disease scores and canopy green cover were integrated over time using the trapezoidal method [32], the named area under disease progress curve (AUDPC), and the healthy area duration (HAD). AUDPC measures the proportion of disease-induced green area loss over time, whilst HAD can be considered a measure of the size of the canopy and the remaining area of healthy photosynthetic tissue [1,10].

All analyses were performed using R version 3.3.0 [33]. Effect of fungicide (Fun), cultivar (Cv), seed treatment (Tr), and their interactions in crop traits or disease (e.g., GY, AUDPC, TGW) were analysed using mixed-effects models. Spatial effects of column and/or subrep were tested selecting the model with lower Bayesian information criterion (BIC) and accounted for as random effects. Assumption of normality and homoscedasticity of variances were checked by QQ-plots and residuals against fitted value plots, respectively. The percentage of senescent tissue (PST) data was $\log_{10}$ transformed to meet normal distribution. Post hoc Fisher's LSD tests were performed to separate significant differences at *p* values $< 0.05$ with *predictmeans* package [34]. *p* values were adjusted to avoid Type I errors (false positives) using the Benjamini–Hochberg correction [35].

Assessments of specific candidate traits that may confer tolerance or escape characteristics were performed using pairwise correlations for each cultivar. Pearson's correlation

between early growth (expressed as LAI) and percentage disease symptoms (PST) was calculated to investigate whether a larger canopy can confer tolerance in pre-stem elongation (early) epidemics. Spearman's correlation was calculated to investigate whether height can be involved in the 'escape' of secondary spread of disease to upper leaves. Specifically, AUDPC accumulated after anthesis in the top four leaves (i.e., flag leaf, leaf 2, leaf 3, and leaf 4 was correlated against the stem elongation rate (in $cm^{-1}$ d) from GS33 (when leaf 3 and leaf 2 emerge) to GS49 coinciding with the rapid stem extension phase.

Disease tolerance of late epidemics (from flag leaf sheath extending onwards) was estimated according to [36] with some modifications. The degree of 'tolerance' was modelled by linear regression as the slope of the relationship between GY and HAD including Cv and Tr as moderator variables. In order to generate sufficient GY-HAD variation for estimation of slopes, data from both sites were pooled, and the effect of fungicide treatments was accounted for as a variation in HAD [36,37]. To validate this approach, the regression slopes were visually checked by specifically ensuring that data were dispersed along the fitted line (i.e., there was no site or fungicide/untreated clusters) before running the model (Figure S4). Spatial effects of column within sites were controlled for by including them as random effects.

## 3. Results

### 3.1. Emergency and Early Growth

Chitosan priming had a positive effect on emergence compared to non-primed seeds, with 22 and 13 more seedlings $m^{-2}$ at Balruddery and Mylnefield respectively at the end of the seedling growth stage, although this increase was only significant at the Balruddery site ($p < 0.01$) (Figure 3). The effect of 'on-farm' seed priming (OSP) was positively related to earliness in emergence (first count event) at Balruddery ($p < 0.01$) but not at Mylnefield. However, this earliness in emergence did not translate into a significantly higher number of seedlings at the end of the seedling growth stage in either of the sites.

Leaf area index (LAI) produced at the advanced tillering stage was estimated using image segmentation. Both sites yielded very similar results with LAI varying by cultivar and seed treatment but with no interaction between them, indicating that the seed treatment effect was similar between the cultivars (Figure 4a,b). KWS Cassia and KWS Tower produced significantly more LAI than SY Venture ($p < 0.001$). Plots sown with non-primed seeds had the greatest LAI overall, whilst those sown with 'on-farm' primed seeds had significantly less LAI at both sites. These results contrast with the positive CHP impact on final emergence, indicating that the effects on emergence did not continue during development up to advanced tillering.

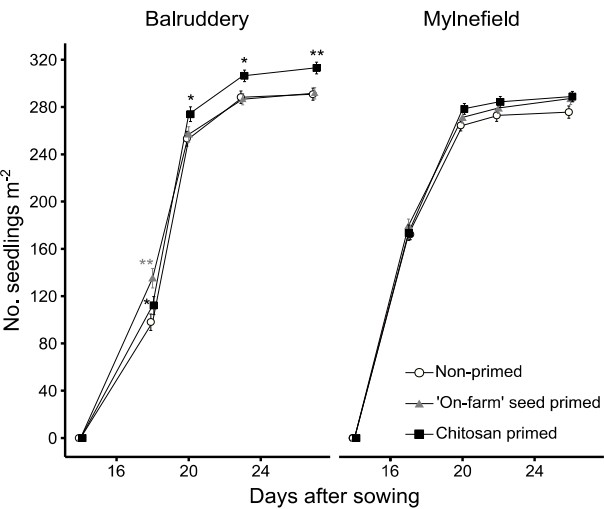

**Figure 3.** Emergence over time. Only seed treatment (Tr) effect is presented, as there was no significant effect of cultivar (Cv) over time. Asterisks denote significant differences (* $p < 0.05$, ** $p < 0.01$) compared to the non-primed control at each time point (LSD test). Each data point represents the mean ($n = 12$); error bars show $\pm$ SE.

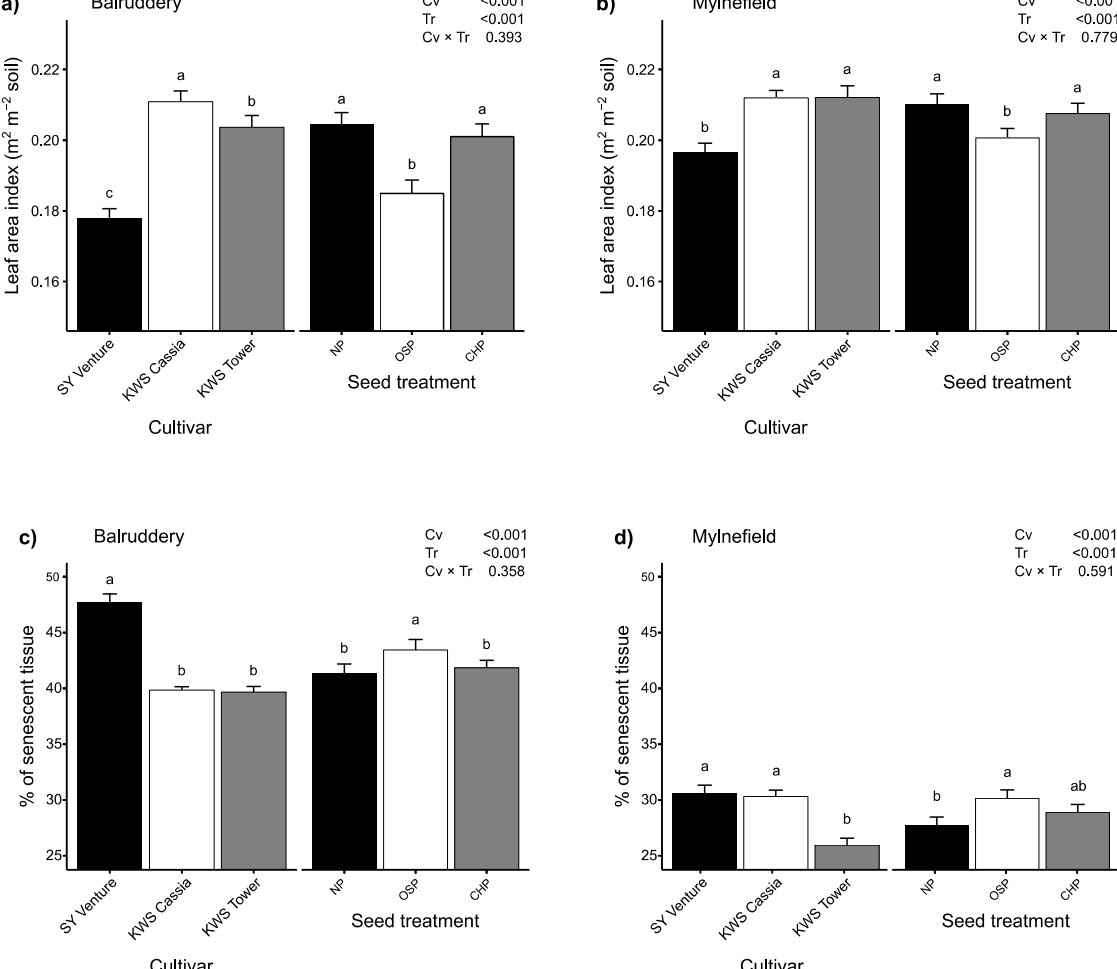

**Figure 4.** Leaf area index (**a,b**) and percentage of senescent tissue (**c,d**) estimated by image segmentation at advanced tillering at Balruddery (**a,c**) and Mylnefield (**b,d**). *p* values from analysis of deviance are for cultivar (Cv) and seed treatment (Tr) effects and the Cv × Tr interaction. NP: non-primed, OSP: 'on-farm' seed primed and CHP: chitosan primed. Bars with different letters are significantly different from each other (LSD test). Error bars show the mean + SE.

### 3.2. Effect of Vigour as a Candidate Trait for Tolerance in Early Epidemics

At the time of image acquisition for image segmentation, both sites were infected with powdery mildew (*Blumeria graminis* f.sp. *hordei*). Most plots at Mylnefield presented discoloured yellow leaves (indicative of the infection depleting the leaf of nutrients) with some grey/brown leaf tips, whilst at Balruddery, damaged tissue was predominantly grey/brown (indicative of an older infection) and also covered with off-white pustules expanding to healthy tissue. Consequently, there was a greater percentage of senescent tissue across cultivars and treatments at Balruddery than at Mylnefield (42% compared with 29%). As for LAI, there were no interactions between factors in any of the trials. The main effects, cultivar (Cv), and seed treatment (Tr) are shown in Figure 4c,d, and post hoc analyses ranked cultivars as SY Venture > KWS Cassia > KWS Tower. Seed treatments showed a similar pattern at both sites with OSP having significantly more senescent tissue than non-primed seeds.

In order to investigate whether crops with larger canopies tend to be more infected during an early disease event, Pearson's correlations between leaf area index and percentage of senescent tissue (PST) were plotted. A consistent negative correlation at both sites for all three cultivars was evident ($p \leq 0.05$), with the relationship being stronger at Mylnefield (Figure 5).

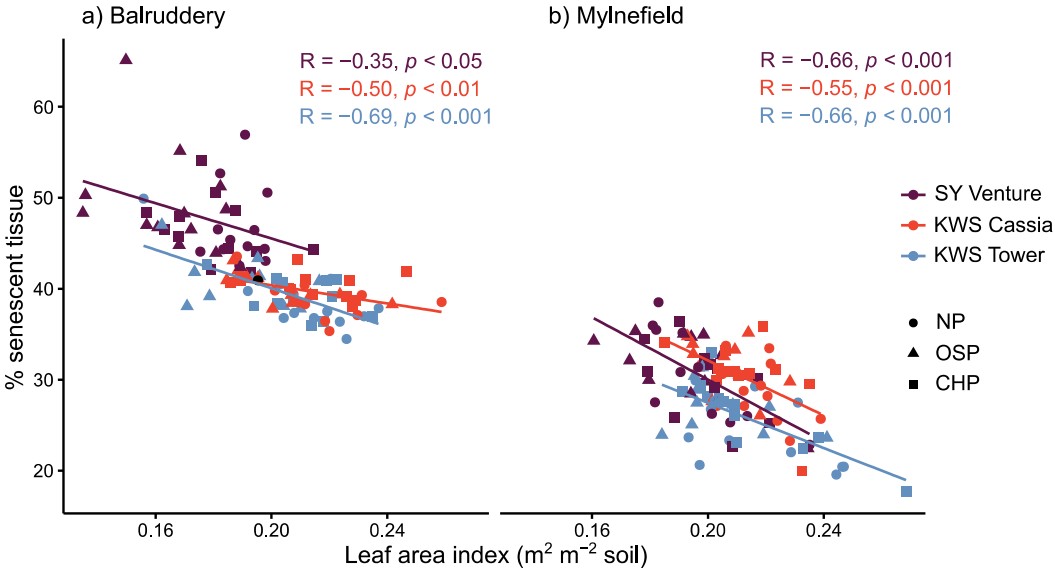

**Figure 5.** Relationship between leaf area index and percentage of senescent tissue at Balruddery (**a**) and Mylnefield (**b**). R: correlation coefficient.

### 3.3. Disease Severity and Resistance

Powdery mildew and rhynchosporium were the only present diseases at measurable levels, although with varying severity and timing between the two sites. Powdery mildew pustules appeared earlier at Balruddery (approximately two months before the start of stem elongation (GS31) and before the first fungicide applications) covering up to 22% of the leaf area (assessed by visual scoring), whilst at Mylnefield, the first pustules appeared about a month later, covering up to 14% of the leaf area (Figures S5 and S6). However, with the appearance of new leaves at the end of stem elongation, powdery mildew infection was reduced to very low levels (<5%) at Balruddery whilst, at Mylnefield, the infection continued to develop and affected parts of leaves 3 and 4 (up to 16% of the total scored leaf area). At Mylnefield, rhynchosporium lesions at traceable levels appeared just before anthesis (GS59) whilst, at Balruddery, there were no rhynchosporium lesions until mid-late anthesis; however, similar levels of severity were recorded by milk development (GS71) at both sites. In terms of visible lesions, fungicide controlled the second increase of powdery

mildew, which occurred when the first awns became visible (GS49) and obscured any rhynchosporium outbreak at both sites.

Area under the disease progress curve (AUDPC) was used to integrate the periodic measurements of disease scores over time as an estimate of disease severity. The main differences in AUDPC were due to the effect of genetic variation (cultivar effect) on both diseases (Table 4). At both sites, KWS Tower was the most resistant cultivar followed by SY Venture and, lastly, by KWS Cassia. At Balruddery, fungicide applications did not significantly reduce powdery mildew AUDPC, largely, because much of the mildew scored corresponded with lesions produced before the first fungicide application rather than connected to the effectiveness of the fungicide controlling the disease. The interaction between fungicide and cultivar for the powdery mildew AUDPC at Mylnefield was due to the fungicide being more effective at controlling powdery mildew in cultivar KWS Cassia compared to SY Venture. However, the interaction between fungicide and cultivar for the rhynchosporium AUDPCs was due to the prevention of rhynchosporium lesions in fungicide-treated plots at both sites. The effect of treatments on AUDPC was only perceptible at Mylnefield for powdery mildew where OSP showed the lowest AUDPC (Table 5). Similarly, the rhynchosporium AUDPC was also the lowest for OSP, although this was not significantly different from NP ($p = 0.27$).

**Table 4.** *p* Values from the analysis of deviance for fungicide (Fun), cultivar (Cv), and treatment (Tr) on the area under disease progress curves (AUDPC) (from GS30 to GS69).

| Site | Term | AUDPC Powdery Mildew | AUDPC Rhynchosporium |
|---|---|---|---|
| Balruddery | Fun | 0.069 | <0.001 |
| | Cv | <0.001 | <0.001 |
| | Tr | 0.954 | 0.136 |
| | Fun × Cv | 0.212 | <0.001 |
| | Fun × Tr | 0.563 | 0.165 |
| | Cv × Tr | 0.870 | 0.243 |
| | Fun × Cv × Tr | 0.701 | 0.285 |
| Mylnefield | Fun | 0.003 | <0.001 |
| | Cv | <0.001 | <0.001 |
| | Tr | 0.040 | 0.189 |
| | Fun × Cv | <0.001 | <0.001 |
| | Fun × Tr | 0.079 | 0.190 |
| | Cv × Tr | 0.981 | 0.458 |
| | Fun × Cv × Tr | 0.943 | 0.457 |

**Table 5.** Effect of seed treatments on the area under disease progress curves (AUDPC). NP: non-primed, OSP: 'on-farm' seed primed, and CHP: chitosan primed. Values in each row followed by different letters differ significantly from each other: LSD test ($p > 0.05$).

| | Tr | | |
|---|---|---|---|
| | **NP** | **OSP** | **CHP** |
| AUDPC powdery mildew | | | |
| Balruddery | 723 [a] | 724 [a] | 725 [a] |
| Mylnefield | 453 [ab] | 427 [b] | 462 [a] |
| AUDPC rhynchosporium * | | | |
| Balruddery | 156 [a] | 189 [a] | 176 [a] |
| Mylnefield | 135 [a] | 115 [a] | 141 [a] |

* values correspond to F0 as there was no AUDPC for rhynchosporium under F1.

### 3.4. Effect of Stem Elongation Rate as a Candidate Trait for Disease 'Escape'

To further explore whether the AUDPC variance found at Mylnefield was to some extent due to the involvement of disease escape mechanisms, a correlation analysis between

rate of stem elongation and AUDPCs from anthesis to grain filling was performed for the plots with no fungicide application (Figure 6). For the case of powdery mildew, this correlation was significantly negative for all cultivars except for SY Venture, showing an average elongation rate above 2.4 cm d$^{-1}$ ($p < 0.01$). However, the same was not applicable for rhynchosporium, as no significant association was found. Stem elongation rate variation was strongly driven by cultivar ($p < 0.001$) and, to a lesser extent, by Tr ($p < 0.05$). OSP had a significantly greater stem elongation rate ($p < 0.05$) (Table S1).

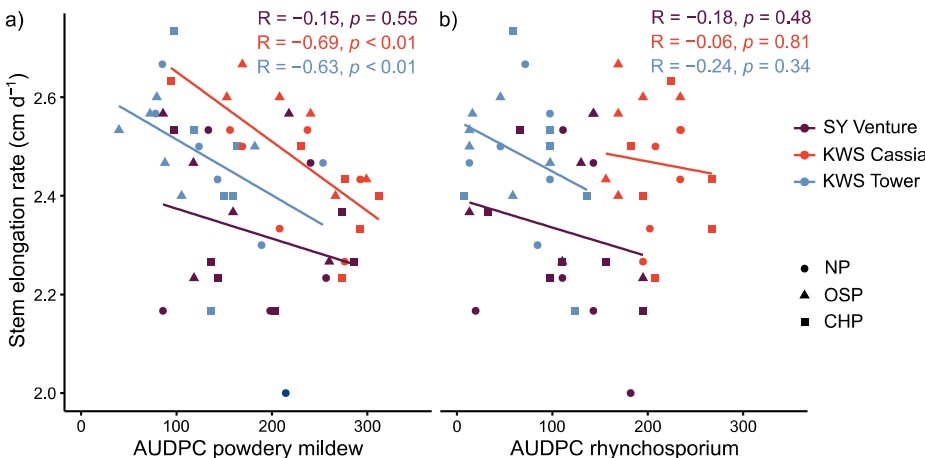

**Figure 6.** Relationship between stem elongation rate from GS33 to GS49 against (**a**) powdery mildew area under disease progress curve (AUDPC) and (**b**) rhynchosporium AUDPC from anthesis in Mylnefield. R: correlation coefficient. NP: non-primed, OSP: 'on-farm' seed primed, and CHP: chitosan primed.

### 3.5. Effects on Yield and Yield Components

Yields were greater at Balruddery (7.73 t ha$^{-1}$) than at Mylnefield (6.68 t ha$^{-1}$), which was mainly attributed to differences in average grain number (13,600 and 11,500 respectively) rather than in TGW (56.68 g vs. 57.8 g respectively). There was a significant grain yield response to fungicide application ($p < 0.05$) with averaged increments across Cv and Tr of 2.02 t ha$^{-1}$ at Balrudery and of 1.15 t ha$^{-1}$ at Mylnefield relative to plots with no fungicide application (Table 6). For sites, this fungicide grain yield response was primarily due to increasing grain number (25 and 13% relative to F0 at Balrudery and Mylnefield respectively) rather than through increments in TGW (4 and 5% respectively). The effect on TGW was significant at Mylnefield ($p < 0.001$), although not at Balruddery ($p = 0.06$). There was a significant interaction between fungicide application and cultivar at Balruddery ($p < 0.05$). Cultivar KWS Tower showed a higher fungicide benefit (2.52 t ha$^{-1}$) compared with SY Venture (1.92 t ha$^{-1}$) or KWS Cassia (1.62 t ha$^{-1}$), despite KWS Cassia being the cultivar with fewer disease lesions. By contrast, there was no interaction between fungicide and cultivar at Mylnefield, indicating that all cultivar genotypes responded similarly to fungicide application. Although seed treatments did not significantly alter yield at Mylnefield, they did at Balruddery. Post hoc analysis showed that grain yield was significantly lower for OSP compared to NP by having a negative impact on grain number, as TGW remained unaffected (Table 7).

**Table 6.** *p* Values from the analysis of deviance for fungicide (Fun), cultivar (Cv), and treatment (Tr) on grain yield (GY), grain number (G no.), and thousand grain weight (TGW).

| Site | Term | GY (t ha$^{-1}$) | G no. (m$^{-2}$) | TGW (g) |
|---|---|---|---|---|
| Balruddery | Fun | 0.010 | 0.009 | 0.060 |
| | Cv | <0.001 | <0.001 | <0.001 |
| | Tr | 0.028 | 0.005 | 0.264 |
| | Fun × Cv | 0.003 | 0.041 | 0.022 |
| | Fun × Tr | 0.193 | 0.281 | 0.237 |
| | Cv × Tr | 0.864 | 0.819 | 0.609 |
| | Fun × Cv × Tr | 0.762 | 0.829 | 0.436 |
| Mylnefield | Fun | 0.015 | 0.045 | <0.001 |
| | Cv | 0.047 | <0.001 | <0.001 |
| | Tr | 0.072 | 0.076 | 0.983 |
| | Fun × Cv | 0.630 | 0.738 | 0.023 |
| | Fun × Tr | 0.103 | 0.243 | 0.103 |
| | Cv × Tr | 0.793 | 0.817 | 0.969 |
| | Fun × Cv × Tr | 0.082 | 0.111 | 0.460 |

**Table 7.** Effect of seed treatment on grain yield (GY), grain number (G no.), and thousand grain weight (TGW). NP: non-primed, OSP: 'on-farm' seed primed, and CHP: chitosan primed. Values between the two farms for each parameter not sharing the same letter differ significantly from each other: LSD test (*p* > 0.05).

| | Tr | | |
|---|---|---|---|
| | **NP** | **OSP** | **CHP** |
| GY (t ha$^{-1}$) | | | |
| Balruddery | 7.89 [a] | 7.54 [b] | 7.77 [ab] |
| Mylnefield | 6.75 [a] | 6.77 [a] | 6.51 [a] |
| G no. (m$^{-2}$) | | | |
| Balruddery | 13,929 [a] | 13,211 [b] | 13,736 [a] |
| Mylnefield | 11,692 [a] | 11,723 [a] | 11,281 [a] |
| TGW (g) | | | |
| Balruddery | 56.5 [a] | 57.0 [a] | 56.5 [a] |
| Mylnefield | 57.8 [a] | 57.8 [a] | 57.8 [a] |

*3.6. Effects on Tolerance in Late Epidemics*

Disease tolerance of late epidemics was represented as the slope of grain yield against HAD after pooling the data from both sites, where the steepness of the slope showed the degree of tolerance (the steeper, the more intolerant). From prior analysis, visual data inspection showed that data were consistent across sites (Figure S4b), whilst fungicide application tended to increase yield over the expected GY-HAD relationship of non-fungicide plots (Figure S4c). There was a significant interaction between HAD and Cv (*p* < 0.05), indicating that the cultivars had different degrees of tolerance (Figure 7a). KWS Tower and SY Venture had similar degrees of tolerance, whilst KWS Cassia was significantly less tolerant than SY Venture (Figure 8a). However, treatments did not have a significant effect on tolerance (*p* = 0.09) (Figure 7b). In general, crops from CHP-treated seeds appeared to have a less steep slope than the non-primed control, but these differences in slope were not significant (Figure 8b).

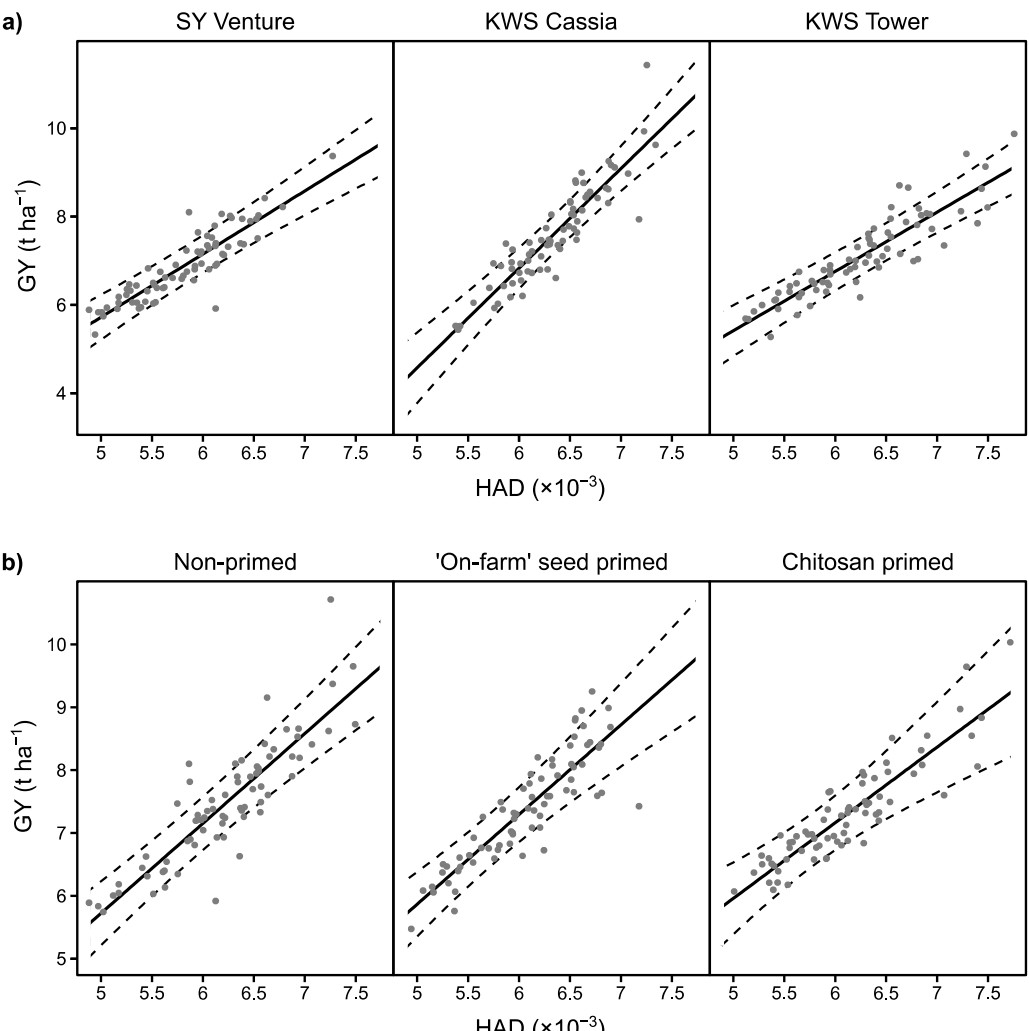

**Figure 7.** Disease tolerance estimated as the slope of grain yield (GY) on healthy area duration (HAD) across sites and fungicide treatments. (**a**) Cultivar effect with all seed treatments pooled together, and (**b**) seed treatment effect with all cultivars pooled together. The solid line represents the regression line, and dashed lines represent 95% confidence intervals.

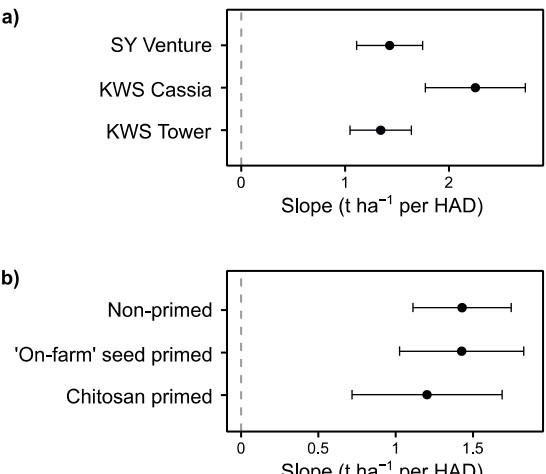

**Figure 8.** Effect sizes for estimated slopes within (**a**) cultivar and (**b**) treatment factor. HAD: healthy area duration. Error bars represent 95% confidence intervals (CI). Effect sizes closer to zero represent greater tolerance. Levels within factor are significantly different from one another when their CIs do not overlap.

## 4. Discussion

'On-farm' seed priming and chitosan seed dressing offer limited scope to control foliar disease in winter barley in temperate agricultural systems, either alone or as a complement to fungicides, regardless of the cultivar of choice. This study has illustrated the varied responses of diseases to conventional management, in particular varietal resistance and fungicides; however, seed treatments do not seem to complement the control of foliar disease.

### 4.1. Induced Resistance Is Hampered under Field Conditions

These trials indicate that disease symptoms are primarily controlled by genetic-mediated resistance (i.e., the cultivar) and, secondarily, by fungicides that can further control the development of disease lesions on new leaves after GS32. However, in general, neither chitosan nor 'on-farm' seed priming further decreased the appearance of lesions, which would have been indicative of induced disease resistance.

The continual interactions between multiple abiotic and biotic agents can compromise the ability of elicitors to further promote host resistance in the field [3,4,38]. Walters et al. [39] ascribed the poor response to elicitors applied to spring barley against powdery mildew and rhynchosporium to the potential for crops already being in an induced state before the application of the elicitors. Stresses such as over-winter cold acclimation, which induces the transcription of a wide array of pathogenesis-related (PR) genes [40], could also mask elicitor-induced disease resistance. Resistance could be induced also by soil microbial communities as demonstrated by Wiese et al. [41]. High organic matter soils showed the lowest powdery mildew infection, whilst the application of the elicitor Acibenzolar-S-methyl (ASM) could only reduce infection in mineral soils [41]. Thus, a more efficient strategy may be to apply elicitors to seeds, rather than in-field applications, to circumvent the impact of abiotic and biotic challenges. This could be especially effective against direct pathogen interaction too, such as with seed- and soil-borne diseases. In this latter respect, some chitosan-based seed treatments have shown promising results as an organic alternative to control seedling blight and foot rot diseases caused by *Fusarium* species in wheat and barley [7,42,43].

### 4.2. Disease Tolerance and Escape in Relation to Crop Traits

Adequate growth and development pre-GS31 may be more important for shoot survival than during the stem extension period, and therefore, it is particularly important for protecting barley crops from early epidemics and for establishing a potential high-yielding crop [1,19]. In this regard, modelling of tolerance traits suggests that a large canopy can be beneficial for tolerating foliar diseases [20]. A large canopy can reduce the impact of disease on growth, as the remaining healthy tissue can compensate potentially for the loss of radiation interception [20]. This mechanism of tolerance is also supported by this study, as a larger canopy tended to have a lower proportion of senescent tissue under moderate and high disease severities of powdery mildew (Figure 5). However, conversely, it is also plausible that a larger canopy could increase the potential for trapping more spores or facilitate the spread to adjacent plants of both wind-borne and splash-spread pathogens such as *Blumeria graminis* and *Rhynchosporium commune*, respectively. The fact that this relationship was strong under moderate severity but less prominent under high severity suggests that this may be possible in the event of very strong epidemics.

In these field trials, seed treatments did not increase canopy size, but rather, 'on-farm' seed priming resulted in reduced early vigour and greater senescent tissue compared to plants sown from untreated seeds (Figure 4). This loss of fitness is difficult to explain, although it is possible that 'on-farm' seed priming washes off important components of seed exudates (exopolysaccharides and organic acids), which are needed to establish beneficial associations with soil microbial communities such as rhizobacteria [44]. This may explain the magnitude of this lessened vigour at Balruddery, which has a richer

environment in terms of microbial communities (as an arable field in a crop rotation) when compared to Mylnefield (in barley monoculture for over 30 years) [45].

Certain traits can constrain the spread of late epidemics to the upper leaves, which contribute the most sink tissue for ear formation and grain filling [1]. In this study, it was found that rapid vertical growth may provide a certain degree of disease escape against powdery mildew but not necessarily to rhynchosporium (Figure 6). Successful attachment of powdery mildew primary germ tube to the leaf surface is enhanced by high humidity [46]. Frequent irrigation created conditions of high humidity at ground level, which in combination with the warm temperatures during late April 2019 provided the ideal microclimate for powdery mildew conidia germination. Thus, it is likely that crops with rapid stem extension developed their upper leaves away from this optimal microclimate and before the pathogen became established, which resulted in fewer powdery mildew lesions post-anthesis. Similarly, height-related traits such as rapid stem elongation, final height, or the distance of the leaf layers to the soil surface can have a negative effect on hemi-biotrophic pathogens such as *Mycosphaerella graminicola* and rhynchosporium in winter wheat and spring barley, respectively [21,47]. However, the relationship between stem elongation and disease lesions may not be so straightforward for rhynchosporium in winter barley, as pathogen load is not only determined by splash-dispersed conidia from lower infected leaves during the early spring precipitation. Earlier infection during the winter may represent another source of pathogen load, as endophytic rhynchosporium can also grow asymptomatically [48].

Whether seed priming can consistently increase stem elongation rate and/or other height-related traits is still unclear. The effect of 'on-farm' seed priming on plant height is either associated with positive effects [14,49,50] or no effect [51,52]. However, it seems clear that the potential effects on phenology are simply the result of quicker establishment that enables a faster growth rate throughout the crop cycle [50] and, thus, exploiting escape benefits will be dependent upon effectively enhancing vigour at establishment.

### 4.3. Overall Tolerance in Relation to Yield as Fitness

Complementary to particular candidate traits, the slope from representing yield against healthy tissue (HAD) can be used to more holistically evaluate tolerance [2]. The approach used in this study for the estimation of tolerance included some modifications of methods previously applied in wheat (e.g., [24,37]). Firstly, pre-anthesis stages were also accounted for in HAD calculation, instead of only post-anthesis. Unlike wheat, barley tiller and spikelet formation are sensitive to variation in radiation interception [53]; hence, this approach allows an integration of this critical period for yield determination into the calculation. Secondly, instead of constructing HAD from the integration of the total planar area of individual sampled plants over time, HAD was calculated from in-field images taken above the canopy over time. This approach is non-destructive and at a field scale provides a better representation of in-field crop architecture. Zenithally taken images give more weight to the upper leaf layer, which intercepts most of the incident radiation, compared with the underlying leaf layers in the calculation, and thus represents a more realistic picture of the impact on radiation interception.

Some caution must be taken when interpreting yield–HAD slopes. Although fungicides are useful to manipulate the disease severity range needed to fit reliable slopes, fungicides can provide yield benefit beyond the ones derived from controlling disease symptoms, which produce some bias [2,54]. In this study, such an effect was apparent at higher HAD (Figure S4). Given that the yield response to fungicide was mostly associated with increasing the grain number per $m^2$, it is likely that HAD is saturated when evaluating dense canopies, underestimating the actual area of healthy photosynthetic tissue. However, it cannot be discounted that deviation over the expected GY–HAD relationship when fungicide is applied may be due to fungicides controlling asymptomatic pathogen infection (whose effect on yield remains largely unknown) or physiological effects derived from its

application [54,55]. Triazoles and strobilurins have been found to alter N partitioning and increase yields [54–56].

There needs to be a compromise between disease tolerance and attainable yield, particularly when disease pressure is low [10,37]. This compromise is illustrated by the less tolerant cultivar (KWS Cassia) having the greatest attainable yield at high HAD, whilst the opposite is true for the most tolerant cultivar (SY Venture) (Figure 7a). This is likely because modern varieties have been bred to perform near optimum radiation use efficiency under fungicide conditions so that a loss in photosynthetically active tissue by disease translates into a more noticeable drop in yield [37]. Although it might be tempting to suggest that chitosan may have some effect on overall tolerance, these differences were marginal and only evident in the most intolerant cultivar when compared to the non-primed control. Taken together, these results of overall tolerance suggest that elicitor seed treatments are only likely to benefit highly susceptible genotypes under high disease pressure.

### 4.4. From Emergence to Yield

'On-farm' seed priming and/or chitosan seed dressing have limited scope for improving winter barley yields and even may result in lower yields. These results are in contrast with spring barley, where both 'on-farm' seed priming and chitosan seed dressing substantially increased grain yields [8]. Yield benefits in spring barley were due to improved emergence and seedling vigour, which led to a greater number of (and more vigorous) tillers being retained for grain filling. However, this was not as effective for enhancing winter barley yields. Although positive effects on emergence density can be gained (chitosan seed dressing seems to provide improved final emergence more consistently than 'on-farm' seed priming), these were not sufficiently high to prevail until advanced tillering.

The mismatch between emergence and canopy cover at advanced tillering in winter crops may be due to their greater plasticity compared to spring barley [57]. The extended canopy formation period (typically from October to the beginning of April) and lower rate of growth imposed by colder temperatures, favours tillering and may allow crops with less initial vigour to catch up. Additionally, the extent of the benefit of earlier emergence may be more limited under typically more humid conditions of autumn-sown crops than those for spring crops. Although crops grown from 'on-farm' primed seeds can attain some earlier emergence, the benefits associated with having moisture already within the seed will be rapidly offset if sown in a damp seedbed. In agreement with these observations, seed priming or chitosan seed dressing have shown limited practical use for enhancing the establishment of winter cereals in temperate climatic zones [11,58,59]. However, there could be considerable benefits for winter cereals grown in semi-arid regions [51,60]. In contrast to temperate zones, winter crops are sown at the beginning of the dry period using the residual water from the rainy season. It is under these circumstances where planting hydrated seeds can make the difference between securing or aborting emergence [61].

### 5. Conclusions

Providing sustainable disease control from seed treatments is attractive for practical and sustainable reasons when compared to spraying fields with fungicides. However, the extent of how seed treatments can complement IPM in conventional temperate agricultural systems seems limited. Inducing resistance from the seed is burdened by continuous interactions with biotic and abiotic elements that offset the expression of induced resistance in field crops. Seed treatments can deliver disease tolerance and escape traits, but these benefits will be conditional upon conferring successful establishment and vigour first. Thus, chitosan-based and 'on-farm' seed priming treatments may be better placed for use with spring crops or in semi-arid agriculture where the added vigour at emergence can more clearly surpass other interactions and facilitate the expression of tolerance and/or escape traits. Finally, a better understanding of the spermosphere and the impact of seed treatments on seed exudates and subsequent germination and interactions with soil micro-

bial communities are also required to design more effective treatments for conventional agriculture.

**Supplementary Materials:** The following are available online at https://www.mdpi.com/article/10.3390/crops1020008/s1, Figure S1: Changes in seed respiration rate during priming 'on-farm' seed priming for each cultivar, Figure S2: Illustration of seedling counting method, Figure S3: Flowchart of image processing for leaf area index (LAI) and percentage of senescent tissue estimation (PST), Figure S4: Visual diagnosis of linearity by relationship between yield (GY) on health area duration (HAD), Figure S5: Area under disease curves at Balruddery. Figure S6. Area under disease curves at Mylnefield, Table S1: Final height and time to 50% GS49 averaged by fungicide and seed treatment.

**Author Contributions:** Conceptualisation, J.C.-R., A.C.N. and R.S.Q.; methodology, J.C.-R., A.C.N. and F.F.-M.; software, J.C.-R. and F.F.-M.; validation, J.C.-R.; formal analysis, J.C.-R.; investigation, J.C.-R.; resources, A.C.N.; data curation, J.C.-R.; writing—original draft preparation, J.C.-R.; writing—review and editing, A.C.N. and R.S.Q.; visualisation, J.C.-R. and F.F.-M.; project administration, A.C.N.; funding acquisition, R.S.Q. All authors have read and agreed to the published version of the manuscript.

**Funding:** This study was funded by Ekhaga Foundation, Sweden (2015-60). The funders had no involvement with the study design; the collection, analysis, and interpretation of data; in the writing of the report, or in the decision to submit the article for publication.

**Data Availability Statement:** The datasets generated during the current study are available in the Stirling Online Repository for Research Data repository.

**Acknowledgments:** The authors would like to thank Shawn Kefauver (Integrative Crop Ecophysiology Group, University of Barcelona, Spain) who kindly provided the CerealScanner plugin and the Scottish Government Strategic Research Programme Theme 2: Productive and Sustainable Land Management and Rural Economies.

**Conflicts of Interest:** The authors declare no conflict of interest. The funders had no role in the design of the study; in the collection, analyses, or interpretation of data; in the writing of the manuscript, or in the decision to publish the results.

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
