# Peer review of "Can ‘On-Farm’ Seed Priming and Chitosan Seed Treatments Induce Host Defences in Winter Barley (Hordeum vulgare L.) under Field Conditions?"

_2673-7655, doi:10.3390/crops1020008_

Round 1
Reviewer 1 Report
The authors tested the impact of alternative strategies aimed at enhancing host defences to improve disease management. The results are clearly present and commented. The paper is interesting and pleasant to read. Despite some negative results (no / little impact of the tested plant defence elicitors), the authors interestingly describe how different strategies can perform more or less well depending on a range of effects / factors. The results give useful elements for agricultural systems seeking to reduce the use of chemical pesticides.
Two main points should be modified before publication. First the methodology used to measure LAI should be justified (see details below). Figures should be referred to in the discussion to simplify reading. Other minor comments are listed below.
L84 “in early the season”: early in the season ?
L85 “for tolerating foliar as it would (…)” : for tolerating foliar diseases ?
L86-87 “traits such as increased height by rapid stem elongation are known to be useful traits for reducing the spread of disease to upper leaves”
I’m not sure rapid stem elongation will systematically cause increased (final) height. However, both traits (fast stem elongation and final height) can contribute to disease escape (in different ways). Replace "by" by "and" or explain the relation between the two traits.
L103 “examine”: examined ?
L216 “Leaf area index (LAI) cover was calculated as the sum of green pixels and yellow 216 pixels and converted to m2 being expressed as m2 of LAI m-2 of soil.”
To me this is ground cover. LAI can be estimated from ground cover in some cases (eg with a particular inclination angle of the camera, see Weiss et al, 2004). The authors should either justify that their measurements actually provide a good estimation of LAI, for eg by citing a reference that validated the protocole by comparing it by the reference protocole by direct destructive measurement of leaf of plants collected in a fixed area. If this is not available, discuss the method in relation to existing works on LAI to justify its relevence or consider changing the name of the variable and explaining its relation with LAI.
Weiss, M., Baret, F., Smith, G.J., Jonckheere, I., Coppin, P., 2004. Review of methods for in situ leaf area index (LAI) determination: Part II. Estimation of LAI, errors and sampling. Agricultural and Forest Meteorology 121, 37–53. https://doi.org/10.1016/j.agrformet.2003.08.001
L316-317 “Plants from non-primed seeds had the greatest LAI overall”
Plants or canopy ? Chitosan impacts the emergence rate so it could have an impact on LAI through number of emerged plants (not LAI / plant). Moreover, reduced density could be compensated by larger contribution of each plant to canopy LAI.
L406 “This grain yield response was primarily due to increasing grain number”: redundant with the first sentence of the paragraph (L401-402)
Discussion: please refer to figures / tables when citing results of the study.
L470-472 “Thus, a more efficient strategy may be to promote elicitor applications to seeds prior to in-field chal-471 lenges.”: I don’t understand this sentence.
L520-521: “The effect of ‘on-farm’ seed priming on plant height is either associated with positive effects [14,48,49] or no effect [50,51].”
Proposition of reformulation: “The effects (...) are either positive (refs) or negative (refs).”
L521-524: “However, it seems clear (…) upon having this prior effect on establishment.” This sentence is unclear to me.
Reviewer 2 Report
Introduction:
Overall I enjoyed the introduction. The conversational tone of the introduction makes it flow well, although the sentence structure could be improved here and there. Consider breaking down long sentences for the sake of readability. Also, a more in-depth discussion on the molecular aspect of how PAMPs alter plant physiology would be helpful. No major flaws.
84-85 Sentence is not clear.
Materials and Methods
The M&M section was clear and understandable. While there appear to be some differences how the field trials were conducted, the authors have made it clear the exact differences between the trials.
Format issue with table 2: The table appears to be too far into the right margin.
Results
Table legends need more information, each table should be able to stand on its own.
Discussion
Double check the use of commas on page 17, and the use of ‘i.e.’. The discussion was fair and proportional to the impact of the data on the overall field.
Summary of manuscript
The manuscript is well written, and the project is clearly outlined. The conclusions the authors took from the results of their experiment was honest and refreshingly humble.
Reviewer 3 Report
It is very important to separate the seed treatment information, which is the focus of the work, from the search for tolerance characteristics. Tolerance characteristics must be studied in a separate experiment for this purpose. External variables not controlled in the experiment may have interfered with the results. The work is relevant in your field. It represents an important contribution to science. It is well written and English is understandable.
Specific comments for authors are in the attached pdf.

Reviewer 4 Report
The study compares the conventional treatment of barley seed priming with that of seeds treated with chitosan for protection, to deliver disease control of winter barley pathogens as sustainable strategy to manage and benefit of the crop. The study meets the analysis requirements applied in the field.
It is mentioned that the selection criteria for the chitosan concentration was 0.5g / L, which means 0.05%, which makes it a low value to cover protection against phytopathogens. There is no bibliographic information that supports this criterion and it would be convenient to mention it in that part of the materials (page 3 line 113).
In the conclusions could add some comment on the cost benefit of this procedure.
Round 2
Reviewer 3 Report
The manuscript improved significantly after review.